

"

# Technical Note: A best-practice approach to calculating the Southern Annular Mode index

Laura Velasquez-Jimenez[1,2,*] and Nerilie J Abram[1,2,3,*]

[1]Research School of Earth Sciences, Australian National University, Canberra ACT 2601, Australia
[2]Australian Centre for Excellence in Antarctic Science, Australian National University, Canberra ACT 2601, Australia
[3]Centre of Excellence for Climate Extremes, Australian National University, Canberra ACT 2601, Australia
[*]These authors contributed equally to this work.

**Correspondence:** Laura Velasquez Jimenez (laura.velasquezjimenez@anu.edu.au)

**Abstract.** The Southern Annular Mode (SAM) strongly influences climate variability in the Southern Hemisphere. The SAM index describes the phase and magnitude of the SAM and can be calculated by measuring the difference in mean sea level pressure (MSLP) between mid- and high-latitudes. This study investigates the effects of calculation methods and data resolution
on the SAM index, and subsequent interpretations of SAM impacts and trends. We show that the normalisation step that is traditionally used in calculating a non-dimensional SAM index leads to substantial differences in the magnitude of the SAM index calculated at different temporal resolutions, and that the equal weighting given to MSLP variability at the mid and high southern latitudes artificially alters temperature and precipitation correlations and the interpretation of climate change trends in the SAM. These issues can be overcome by instead using a dimensional formulation of the SAM based on MSLP anomalies,
resulting in consistent scaling and variability of the SAM index calculated at daily, monthly and annual data resolutions. The dimensional version of the SAM index has improved representation of SAM impacts in the high southern latitudes, including the asymmetric (zonal wave-3) component of MSLP variability, whereas the increased weighting given to mid-latitude MSLP variability in the non-dimensional SAM incorporates a stronger component of tropical climate variability that is not directly associated with SAM variability. We conclude that a best-practice approach of calculating the SAM index as a dimensional
index derived from MSLP anomalies would aid consistency across climate studies and avoid potential ambiguity in the SAM index, including SAM index reconstructions from paleoclimate data, and enable more consistent interpretations of SAM trends and impacts.

## 1   Introduction

The Southern Annular Mode (SAM) is the leading mode of atmospheric variability in the extratropical Southern Hemisphere.
The SAM describes changes in the strength and position of the westerly wind belt and associated storm tracks, and can be characterised through the difference in zonal mean sea level pressure (MSLP) between the southern mid-latitudes and Antarctica (Thompson and Wallace, 2000; Marshall, 2003). A positive SAM is characterised by positive pressure anomalies at mid-





latitudes and negative pressure anomalies over Antarctica (Fig 1; Marshall, 2003). These variations in the latitudinal pressure

gradient have been found to influence temperature and precipitation across the Southern Hemisphere, and also interact with

other major modes of climate variability. For example, a positive SAM has been associated with decreases in precipitation and

positive temperature anomalies in southeast South America often as a result of interactions with El Niño-Southern Oscillation

(Silvestri and Vera, 2003; Vera and Osman, 2018). In South Africa, a positive SAM is associated with a decrease in rainfall

during winter and spring related to a shift in the polar jet (Reason and Rouault, 2005). In Australia, a positive SAM during

winter is linked to reduced precipitation in southern parts of the country, while a negative SAM in summer can lead to re-

duced rainfall and elevated temperature and bushfire risk in parts of eastern Australia (e.g., Meneghini et al., 2007; Mariani

and Fletcher, 2016; Lim et al., 2019; Abram et al., 2021). While in New Zealand, a positive SAM is linked to a decrease in

precipitation and an increase in temperature due to weakened westerly winds passing over the islands (Kidston et al., 2009).

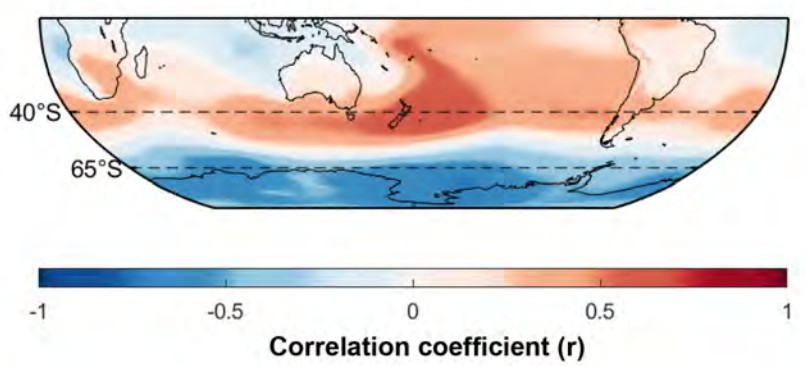

**Figure 1.** Spatial correlation of SAM index to mean sea level pressure (MSLP) in the Southern Hemisphere. SAM index was calculated from annually means (January-December; 1950-2022, ERA5 data) using the difference in zonal MSLP at 40°S and 65°S (dashed lines).

The phase and magnitude of SAM variability is described by the SAM index. Two methods are commonly used to calculate

the SAM index. The first method is based on gridded data such as atmospheric reanalysis (e.g. ERA5) or climate model output,

and breaks down extra tropical Southern Hemisphere atmospheric pressure data into orthogonal spatial patterns expressed by

Empirical Orthogonal Functions (EOF). The first EOF explains the leading mode of Southern Hemisphere variability and its

time series represents the SAM Index (Mo, 2000; Fogt and Bromwich, 2006). Recent advances in the application of the EOF

method to describe the SAM include approaches to separate the zonally symmetric component of SAM variability from the

asymmetric component of variability associated with the zonal wave-3 pattern (Goyal et al., 2022; Campitelli et al., 2022). The

second method for calculating the SAM index uses the difference in the normalised zonal mean sea level pressure (MSLP)

between 40°S and 65°S (Fig. 2). By this method the SAM Index can be calculated using gridded products (Gong and Wang,

1999) or instrumental records of MSLP from observing stations located in the southern mid-latitudes and around coastal



Antarctica (Marshall, 2003). It is this second method of calculating the SAM index that is the focus of the assessment carried out in this study.

Instrumental climate measurements are sparse across the Southern Hemisphere, and particularly in Antarctica. This generally limits a reliable long term understanding of SAM variability from observations and reanalysis products to the time since 1957 (Marshall, 2003; Barrucand et al., 2018; Marshall et al., 2022), although some longer reconstructions based on observations have also been developed back to the late 19th century (Jones et al., 2009; Visbeck, 2009). Over this historical period there has been a significant positive trend in the SAM, particularly in the summer season, associated with stratospheric ozone loss as

well as rising atmospheric greenhouse gases (Thompson and Solomon, 2002; Fogt and Marshall, 2020). This trend is expected to continue in all seasons during the 21st century as climate continues to warm due to ongoing anthropogenic greenhouse gas emissions, but with a temporary pause in summer trends due to the opposing influence of stratospheric ozone recovery (Thompson et al., 2011; Goyal et al., 2019; Banerjee et al., 2020)

Longer-term reconstructions of the SAM have been developed using paleoclimate proxy records (e.g., ice cores, tree rings

and corals, etc) and multiple reconstructions for the last millennium have been produced (e.g., Villalba et al., 2012; Abram et al., 2014; Dätwyler et al., 2018; King et al., 2023). These long-term reconstructions show similar trends in the SAM index, however, they display different magnitudes of reconstructed SAM variability. Although variability between reconstructions could be due to differences in reconstruction methods and the networks of proxy data used, Wright et al. (2022) instead found that differences in magnitude between the Abram et al. (2014) and Dätwyler et al. (2018) reconstructions were explained by the

data resolution used to calculate the instrumental SAM index. Dätwyler et al. (2018) trained their reconstruction to an annual SAM Index calculated from monthly MSLP data, while Abram et al. (2014) used the annual SAM Index from annual MSLP data as their reconstruction target. The difference in magnitude of the annual SAM index in instrumental data calculated by these alternate methods accounts for the apparently larger (though dimensionless) magnitude of SAM variability during the last millennium in the Abram et al. (2014) reconstruction compared with the Dätwyler et al. (2018) reconstruction (Wright

et al., 2022). This discrepancy highlights the importance of understanding the impact of methodology in reconstructing the SAM index from observational data.

It has previously been shown that differences between the method (e.g. EOF or zonal difference index methods), variable (e.g. pressure level) or source data (e.g. gridded reanalysis or station observations) results in sometimes marked differences between available observational SAM indices, despite these indices all representing the same physical process (Ho et al., 2012).

However, it is not known how methodological choices within a single method, variable and data source might also have the potential to influence the results of SAM studies. To date, a best-practise data resolution to use when calculating the SAM index has not been established, and various versions constructed using different resolutions and orders of operation are made available for the research community to use (e.g. http://www.nerc-bas.ac.uk/icd/gjma/sam.html). It also remains unexplored if the choice to normalise zonal MSLP data prior to calculating the latitudinal difference in pressure anomalies (Gong and Wang,

1999; Marshall, 2003) could influence the assessment of past and future SAM changes, or the climate impacts that SAM causes in different parts of the Southern Hemisphere.



Here, we calculate historical SAM indices using daily, monthly and annual averages of zonal MSLP data, and using dimensional and non-dimensional formulations of the SAM index. We explore differences between the SAM indices, and the reasons why methodological choices introduce these differences, as well as the potential implications when analysing the spatial correlation of SAM variability with temperature and precipitation impacts. Additionally, we also explore the influence of methods on the interpretation of SAM trends in projections of climate change during the 21st century. We conclude by making recommendations for a best-practice approach to calculating the SAM index that avoids potential biases introduced by methodology.

## 2  Methods

We use the ECMWF (European Centre for Medium-Range Weather Forecasts) Reanalysis v5 (ERA5) gridded data for our study (Hersbach et al., 2020). ERA5 reanalysis data is currently available from 1950. Of the available reanalysis products, ERA5 has been shown to best reproduce Antarctic surface temperature and SAM relationships prior to the satellite era (Marshall et al., 2022).

Daily resolution MSLP data in ERA5 for latitudes 40°S and 65°S were sourced from the KNMI Climate Explorer tool (Trouet and Van Oldenborgh, 2013). From daily ERA5 data, the daily, monthly and annual means of zonal MSLP were calculated. SAM Indices were then calculated for these three different data resolutions (Fig 2).

Following the approach of Gong and Wang (1999), the SAM Index was first calculated using the equation:

$$SAM = P^*_{40°S} - P^*_{65°S} \tag{1}$$

where P*$_{40°S}$ and P*$_{65°S}$ are the normalised zonal MSLP at 40°S and 65°S, respectively.

Data was normalised relative to a 1961–1990 reference interval. Briefly, this involves subtracting the mean of the reference interval from the time series, and then dividing the time series by the reference interval standard deviation. The SAM index was then calculated by subtracting the normalised zonal MSLP values at 65°S from the normalised zonal MSLP values at 40°S (Fig. 2). The normalisation step removes units from the MSLP data, and consequently also from the resultant SAM index, and so we refer to this as the non-dimensional SAM index.

A dimensional SAM index in hPa pressure units was also calculated (Fig. 2). This followed the same equation and method as above, but in this case P*$_{40°S}$ and P*$_{65°S}$ are the zonal MSLP anomalies at 40°S and 65°S. Specifically, for the dimensional SAM index the zonal MSLP anomalies are calculated relative to the 1961–1990 reference interval mean without dividing by the reference interval standard deviation.

The relationship between daily, monthly and annual SAM index methods was then investigated by calculating an annual mean SAM from the daily and monthly indices (Fig. 2). The annual SAM values derived from the different resolution SAM indices were then compared by a correlation coefficient (r) and by examining the gradient between different methods of calculating the SAM Index. The spatial correlation of each SAM index at each data resolution with ERA5 gridded data for 2m air temperature




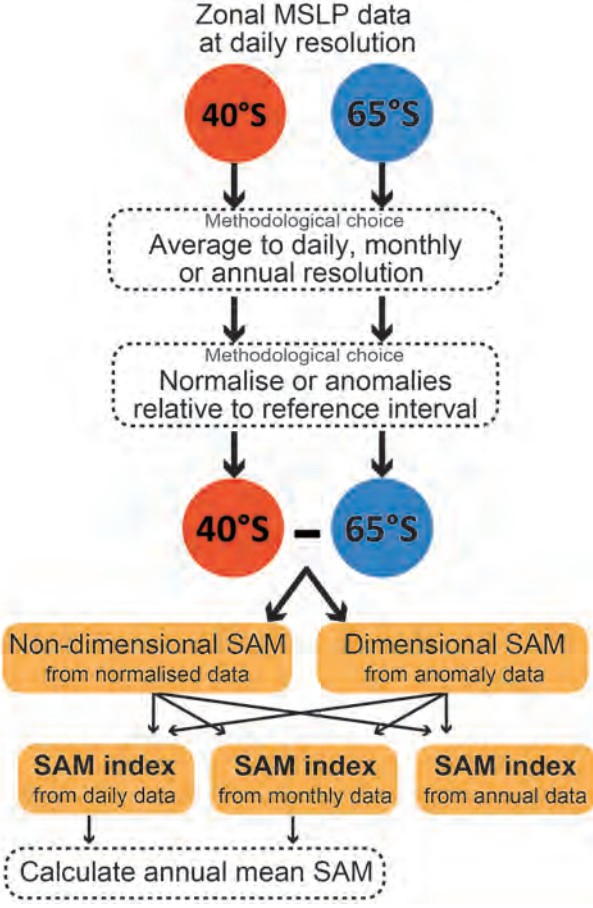

**Figure 2.** Methodological choices explored in this study by calculating dimensional and non-dimensional SAM indices from different data resolutions.

and precipitation was also examined to test the influence of methodological choices on detection and interpretation of the SAM's climate impacts.

To illustrate the impact that methodological choices could have on the interpretation of future SAM changes we also test climate
model output from 1850 to 2100. To illustrate the effect of methodological choices we use output from the CSIRO ACCESS-CM2 model prepared for CMIP6 (Dix et al., 2019). A full assessment of future SAM changes would require a more thorough analysis across the ensemble of CMIP6 models, as done for example in Goyal et al. (2021), but our purpose in this study is to simply illustrate the potential impact of methodological choices on such assessments. MSLP outputs from the ACCESS-CM2 model were sourced from the "very high" and "low" emission scenarios for future climate change (SSP5-8.5 and SSP1-2.6,
respectively) in order to best identify the range of influences that methodological choice could have on assessing SAM changes in a warming climate. As the output from these global climate model simulations are routinely reported at monthly mean





resolution, only monthly and annual mean SAM indices were calculated for the future projections. Both non-dimensional and dimensional SAM indices were calculated from the climate model output, relative to a 1961-1990 reference interval.

All data analysis were carried out using MATLAB R2022b software. This included using the M-map package and the Climate
Data Toolbox for producing the analyses and maps presented in this study (Greene et al., 2019; Pawlowicz, 2020).

## 3 Results

### 3.1 SAM index characteristics

Data resolution strongly influences the magnitude of the non-dimensional SAM index (Fig. 3a). While the pattern of interannual variability of the non-dimensional SAM is very similar for all data resolutions (as demonstrated by r values exceeding 0.99;
Fig. 3b-c), the magnitude of interannual variability of the non-dimensional SAM derived from monthly data is 1.4 times larger than the non-dimensional SAM derived from daily data (Fig. 3b). Similarly, the magnitude of the annual non-dimensional SAM index calculated from annual means is 3.1 times larger than the non-dimensional SAM derived from monthly data (Fig. 3c) and 4.4 times higher than the annual SAM derived from daily data. This finding is consistent with the recalculation performed by Wright et al. (2022) where the SAM index calculated from annual MSLP data displayed a higher variability than annual
means derived from a monthly SAM index.

Differences in magnitude of the non-dimensional SAM index are caused by a progressive decrease in standard deviation as MSLP data is averaged over longer time periods (Table 1). This means that the normalisation of daily MSLP data removes a larger magnitude of variability than normalisation of monthly MSLP data, and even more so when comparing to normalisation of annual resolution MSLP data. Comparison of the reference interval MSLP standard deviations between the different data
resolutions (Table 1) gives similar ratios to the slopes between the annual mean SAM values derived from different resolution SAM indices in Figure 3a-c. For example, the normalisation step in calculating the SAM index removes a 3.3 times greater magnitude of MSLP variability at 40°S for monthly resolution data compared to annual mean data (standard deviations of 1.694 and 0.509 hPa, respectively; Table), and 3 times more variability at 65°S (standard deviations of 4.025 and 1.355 hPa, respectively; Table 1). This results in the 3.1 times greater magnitude of interannual SAM variability calculated from annual
data relative to monthly data when using the normalisation method to calculate a non-dimensional SAM index (Fig. 3c).

Differences in the magnitude of the SAM index are overcome when a dimensional SAM index is instead calculated. The annual mean dimensional SAM values calculated from daily, monthly and annual resolution MSLP data all display the same phase and magnitude of interannual variability over time (Fig 3d). This highlights how the normalisation step used in calculating the non-dimensional SAM index can introduce ambiguity into SAM studies, but also how this ambiguity can be avoided by
retaining the native pressure units in the SAM index.

Our findings also demonstrate that a dimensional SAM index can be reliably calculated from low resolution MSLP data. Physically, it is the instantaneous difference in pressure between the mid and high southern latitudes that represents the processes of

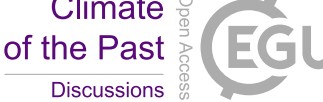



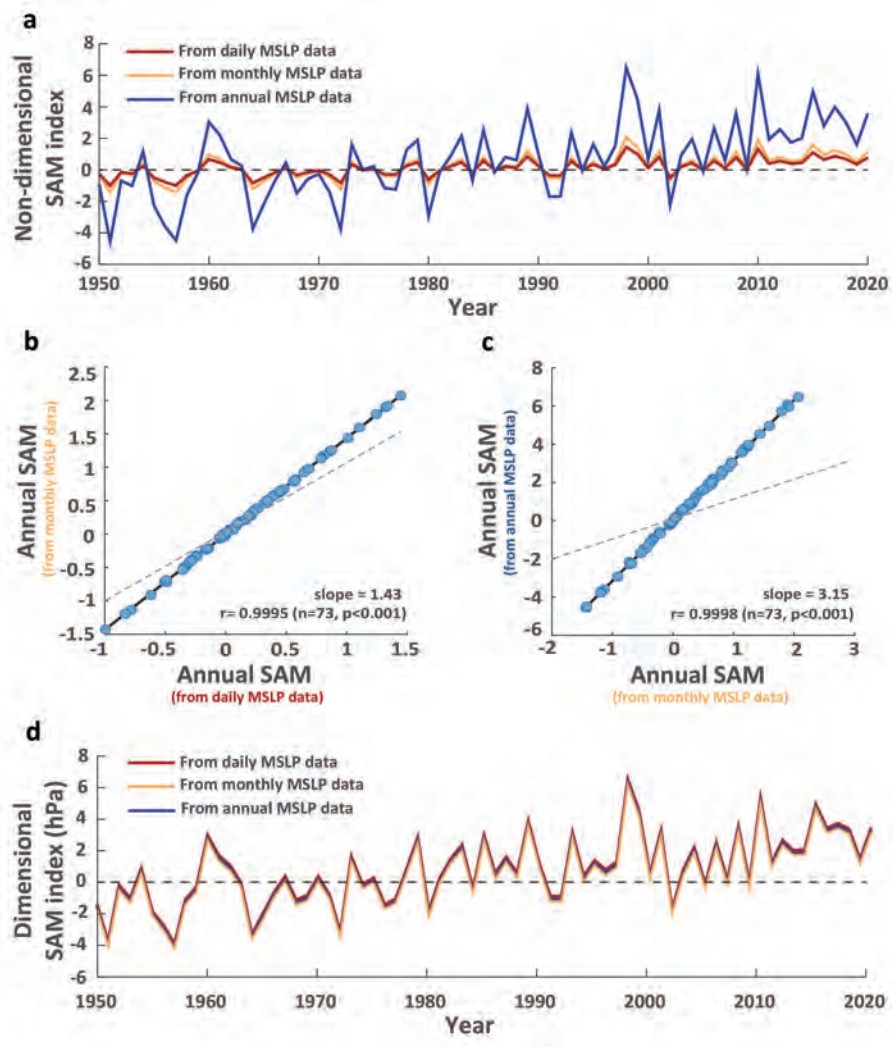

**Figure 3.** Annual mean SAM values calculated by different methodological choices. a. Comparison of annual non-dimensional SAM values calculated from daily (red), monthly (orange) and annual (blue) MSLP data. b. Relationship between the annual non-dimensional SAM values calculated from daily and month resolution MSLP data. Dashed line represents 1:1 slope c. Relationship between the annual non-dimensional SAM values calculated from monthly and annual resolution MSLP data. Dashed line represents 1:1 slope. d. Comparison of annual dimensional SAM values calculated from daily (red), monthly (orange) and annual (blue) MSLP data.

atmospheric SAM variability (Baldwin, 2001), and so daily resolution data might be assumed to retain a more pure measure of the SAM index. However, our findings using different resolutions of MSLP data show that the interannual trends and variability of the dimensional SAM are consistently captured using daily, monthly or annually averaged zonal MSLP anomalies (Fig. 3d).





Beyond scaling, there are additional (though small) year-to-year differences in the interannual variability and trends of the SAM when comparing dimensional and non-dimensional calculations of the SAM index. These differences are evident when comparing annual SAM values calculated as a dimensional or non-dimensional index from annual MSLP data (Fig. 4), and are similarly evident when comparing the variability of dimensional and non-dimensional SAM indices calculated from monthly
MSLP data or from daily MSLP data (not shown).

These differences in year-to-year variability and trends can again be explained as an artefact introduced by the normalisation step when calculating the non-dimensional SAM index. By normalising the zonal MSLP data before calculating the zonal difference, an identical weighting is given to pressure variability in the mid and high latitudes in the calculation of the non-dimensional SAM index. However, the magnitude of MSLP variability is consistently larger at 65°S compared with 40°S
(Table 1). At daily resolution the magnitude of reference interval variability at 65°S is 2.22 times larger than the variability at 40°S (standard deviations of 5.597 hPa and 2.524 hPa, respectively), and at annual resolution variability at 65°S is 2.66 times larger than at 40°S (standard deviations of 1.355 hPa and 0.509 hPa, respectively). Likewise, the long-term trends in MSLP are amplified at 65°S (-0.50 hPa/decade from 1950-2022) compared to the MSLP trends at 40°S (0.18 hPa/decade). These differences suggest that the equal weighting of these latitudinal zones that is routinely applied in calculating the non-
dimensional SAM index may not be justified, and could artificially alter the interpretation of SAM variability, trends and impacts.

**Table 1.** Characteristics of MSLP variability during the 1961-1990 reference interval for the zonal MSLP data used to calculate the SAM index at different resolutions.

| Data resolution | 40°S standard deviation (hPa) | 65°S standard deviation (hPa) |
|---|---|---|
| Daily | 2.524 | 5.597 |
| Monthly | 1.694 | 4.025 |
| Annual | 0.509 | 1.355 |

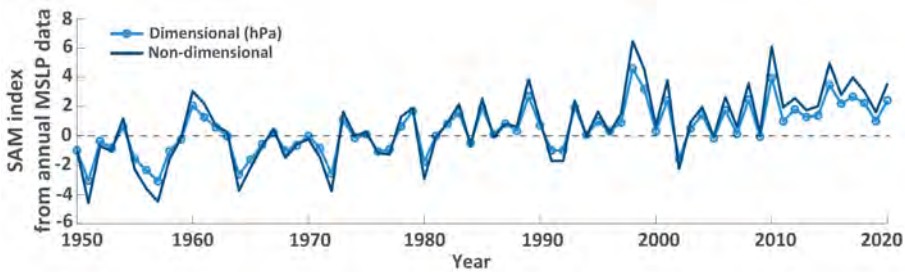

**Figure 4.** Comparison of interannual variability and trends from dimensional and non-dimensional annual SAM values calculated from annual MSLP data.





### 3.2 SAM impacts

Spatial correlation analysis shows that the SAM index is correlated with Southern Hemisphere temperature variability, with similar broad-scale patterns across SAM index data resolutions and calculation methods (Fig. 5). In general, all formulations
of the SAM indices produce negative correlations with annual mean temperature anomalies over the Antarctic continent, and positive correlations over the Antarctic Peninsula and southern South America, over the southern Indian Ocean, and over the Maritime continent extending into the eastern tropical Indian Ocean, the Coral Sea and the Tasman Sea. However, beyond these broadly consistent patterns we demonstrate that the methodology used to construct the SAM index does alter the strength of temperature correlations in some locations.

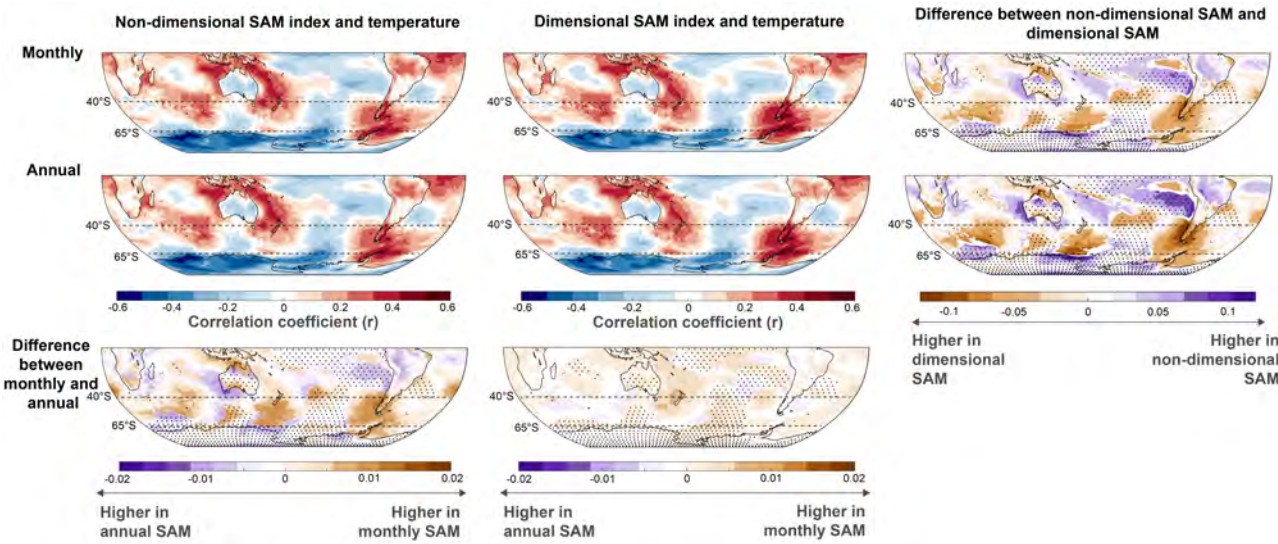

**Figure 5.** Spatial correlation of annual SAM values with ERA5 2m air temperature in the Southern Hemisphere (January-December averages over 1950-2022). Comparisons are shown for differences in SAM indices derived from monthly (top row) and annual (middle row) MSLP data, and for non-dimensional SAM indices (left column) and dimensional SAM indices (middle column). Also shown are the differences in spatial correlation values based on MSLP data resolution (bottom row) and for dimensional versus non-dimensional SAM indices (right column). In these correlation difference plots the shading represents differences between methods and data resolution while stippling indicates regions of negative spatial correlations. Consistent findings are also produced comparing annual temperature correlations for SAM indices derived from daily and annual MSLP data (Fig. A1).

Comparing the correlations produced by dimensional versus non-dimensional formulations of the SAM index (i.e. comparing along rows in Fig. 5) clear spatial characteristics in correlation differences area evident. Generally, correlation strength in the region between 40°S and 65°S is stronger for the dimensional SAM than it is for the non-dimensional SAM. These differences in correlation strength show three distinct nodes across the Southern Ocean and Drake Passage suggesting that the dimensional SAM index better includes the asymmetric (zonal wave-3) component of SAM variability. In contrast areas north of 40°S more
commonly have stronger correlations with the non-dimensional SAM index. It is expected that this is because the normalisation



of the non-dimensional SAM index artificially increases the weighting of MSLP variability at 40°S (relative to MSLP variability at 65°S). This would emphasise the temperature effects of pressure variability in the mid-latitudes as well as their interactions with tropical circulation such as the Hadley and Walker circulation cells.

Two important features are found when comparing the annual temperature correlations produced by different resolutions of the
SAM index (i.e. comparing down columns in Fig. 5). Firstly, differences in resolution of the non-dimensional SAM produce similar spatial patterns of correlation differences as are seen in the comparision between dimensional and non-dimensional SAM indices. Specifically, the non-dimensional SAM generated from monthly resolution MSLP data has stronger correlations with interannual temperature variability in the region between 40°S and 65°S, including showing improved correlation with the zonal wave-3 pattern. The non-dimensional SAM generated from annual resolution MSLP data has generally stronger
correlations with interannual temperature variability north of 40°S. These differences are emphasised even further in comparing annual temperature correlations with the non-dimensional SAM generated from daily versus annual MSLP data (Fig. A1). This is again explainable through the increasingly strong weighting that is given to pressure variability at 40°S relative to variability at 65°S as MSLP data resolution is reduced in calculating the non-dimensional SAM (Table 1). However, the other important finding that is evident in this analysis is that the spatial differences in correlation strength associated with MSLP data resolution
can be avoided altogether by using a dimensional SAM index (middle column of Fig. 5).

Similar findings come from examining the correlation of annual precipitation with the various methodological choices for calculating the SAM index (Fig. 6). The primary correlation patterns with precipitation show broad agreement across methods. Positive mean annual SAM anomalies are associated with latitudinal bands of increased precipitation near the Antarctic coast (including over the Antarctic Peninsula) and a band of decreased precipitation across the mid-latitudes. This represents the
southward shift of the westerly winds and associated storm tracks when the SAM is in its positive phase. Other regions demonstrating positive mean annual precipitation associated with positive SAM anomalies include the Maritime Continent including the eastern tropical Indian Ocean and eastern Australia and the tropical eastern and central Pacific. Negative mean annual precipitation anomalies are also seen over West Antarctica in response to positive SAM phases.

Beyond these broad similarities in SAM correlations with precipitation, we do again identify regions where methodological
choices alter the correlation results produced (Fig. 6; Fig A2). Correlations with interannual precipitation variability near 65°S, and particularly over the Antarctic Peninsula, are generally stronger for higher resolution versions of the non-dimensional SAM index, and for all resolutions of the dimensional SAM index. Conversely, correlations with interannual precipitation variability near 40°S, and specifically south of Australia, over the south island of New Zealand and west of Chile, are stronger for lower resolution versions of the non-dimensional SAM, and for the non-dimensional SAM compared with the dimensional SAM.
These formulations of the SAM index also show stronger precipitation anomalies over parts of the tropics including northern Australia and the Amazon region, indicating the stronger representation of tropical-to-mid-latitude atmospheric circulation in these versions of the SAM index that give increased weighting to pressure anomalies at 40°S. In other words, it is these regions where methodological choices in constructing the SAM index will have the most impact on the interpretation of the SAM's influence on annual mean precipitation.





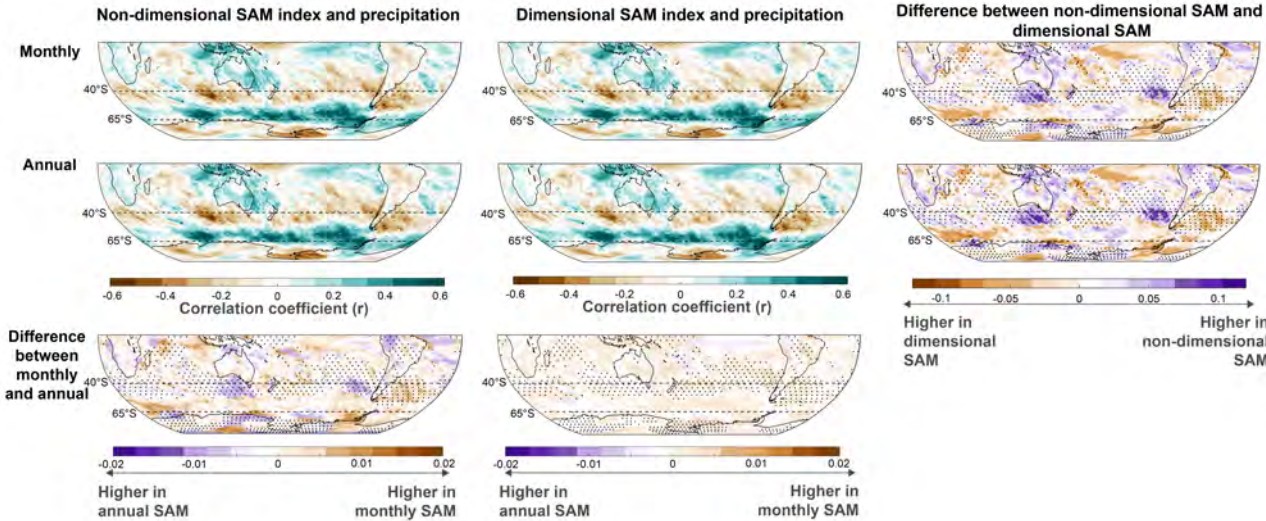

**Figure 6.** Spatial correlation of annual SAM values with ERA5 precipitation in the Southern Hemisphere. (January-December averages over 1950-2022). Comparisons are shown for differences in SAM indices derived from monthly (top row) and annual (middle row) MSLP data, and for non-dimensional SAM indices (left column) and dimensional SAM indices (middle column). Also shown are the differences in spatial correlation values based on MSLP data resolution (bottom row) and for dimensional versus non-dimensional SAM indices (right column). In these correlation difference plots the shading represents differences between methods and data resolution while stippling indicates regions of negative spatial correlations. Consistent findings are also produced comparing annual precipitation correlations for SAM indices derived from daily and annual MSLP data (Fig. A2).

We note that these comparisons are shown for mean annual precipitation and SAM anomalies, but it is well established that the impacts of SAM on precipitation vary by season (Fogt and Marshall, 2020). Because of this, the impacts of methodological choices in assessing the SAM's precipitation impacts at a seasonal scale may result in different regions where those method-ological choices alter correlation strength. However we expect that our general conclusions would remain the same at the seasonal scale, including that a dimensional version of the SAM index would produce correlation results that are unaffected by 220 choices in the resolution of zonal MSLP data used to construct the SAM index.

### 3.3 SAM trends

Finally, we look at how methodological choices in constructing the SAM index could alter the interpretation of SAM changes in a warming world. During the historical period the differences in interannual variability of annual SAM values produced by dimensional or non-dimensional SAM indices are detectable but small (Fig. 4). However, as the response to human-caused 225 climate warming develops, the magnitude of SAM trends relative to the magnitude of historical variability show increasing differences between different methodological versions of the SAM index (Fig. 7).





**Figure 7.** Example of future scenario SAM indices based on different calculation methods. a. Comparison of low emissions future scenario (SSP1-2.6) based on dimensional (purple) and non-dimensional (green) SAM indices calculated from annual MSLP data for 1850-2100. Thick lines show 50-yr moving averages. Reference interval used for calculating the SAM indices is 1961-1990. b. As in a, but for a very high emissions future scenario (SSP5-8.5)

Long-term climate change trends are stronger in the non-dimensional SAM compared to the dimensional SAM, relative to historical interannual variability (Fig. 7). This difference will affect interpretations of time of emergence (Hawkins et al., 2020), which assess when a long-term climate trend (signal) emerges above the amplitude of historical climate variability (noise) resulting in climate conditions that are beyond the range of historical experience. For example, under a future with very high greenhouse gas emissions (SSP5-8.5) the climate change signal on the SAM index (as assessed by a 50-year moving average) emerges above the 1 standard deviation historical (1850-1949) noise level by 2025, and above the 2 standard deviation historical noise level by 2091, in a non-dimensional formulation of the SAM. In contrast, for the dimensional SAM there is emergence above the 1 standard deviation level by 2031, but no emergence occurs above the 2 standard deviation level during

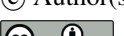



the 21st century. Likewise, under a low greenhouse gas emissions scenario (SSP1-2.6) there is emergence of the climate change
signal for the non-dimensional SAM between 2063 and 2086, but emergence is not detected at any time during the 21st century
for the dimensional SAM.

This finding illustrates how methodological differences in calculating the SAM index have the potential to alter interpretations
of human-caused climate impacts on the SAM. Our findings suggest that the normalisation associated with a non-dimensional
SAM index may lead to assessments that the SAM has emerged outside of the range of historical experience sooner than would
be determined based on a dimensional SAM. We emphasise that this is only an illustrative example based on a single climate
model, but it does demonstrate the potential for methodological choices to influence the interpretation of SAM trends between
different studies.

## 4 Discussion and Conclusions

Our results allow us to make recommendations for a best-practice approach to calculating the SAM index to enable consistency
across climate studies. The traditionally used (non-dimensional) SAM index (Gong and Wang, 1999; Marshall, 2003) involves
normalising zonal MSLP data before calculating the latitudinal MSLP difference that defines the SAM. It isn't clear why
the choice to normalise zonal MSLP data was originally made, although it is possible that this was to facilitate comparisons
with EOF-based methods of defining the SAM that produce non-dimensional principal components (Gong and Wang, 1999;
Baldwin, 2001), or because of potential spurious trends in early MSLP data in the Antarctic region (Baldwin, 2001; Marshall,
2003).

We find that the normalisation step involved in the traditionally used (non-dimensional) SAM index has the potential to intro-
duce multiple biases in climate studies. Firstly, the magnitude of the non-dimensional SAM index value varies substantially
based on the temporal resolution of zonal MSLP data used to construct the SAM index (Fig 3a-c). Because the index produced
by this method is dimensionless these differences are hard to trace when SAM indices are then applied in climate research, and
there are examples where this has then resulted in seemingly large differences in the magnitude of paleoclimate reconstructions
of the SAM (Wright et al., 2022). The normalisation step in calculating the non-dimensional SAM also gives equal weighting
to MSLP variability and trends in the mid and high latitudes. However, the magnitude of MSLP variability and trends are
substantially larger at 65°S compared to 40°S (Table 1). The effect of equally weighting MSLP anomalies at 40°S and 65°S
results in biases in correlations with temperature and rainfall data that could alter the interpretation and attribution of SAM
impacts in some regions. This includes generally reducing SAM correlations with temperature and precipitation variability in
the high southern latitudes, and giving enhanced influence to the impacts of mid-latitude pressure anomalies and their links to
tropical atmospheric circulation (Figs. 5 and 6). Furthermore, the non-dimensional SAM index displays stronger future climate
change trends relative to the magnitude of historical variability. Because of this the SAM would be assessed to emerge above
historical experience sooner this century using a non-dimensional SAM index compared with a dimensional index (Fig. 7).





These problems are overcome when using a dimensional version of the SAM based on zonal MSLP anomalies rather than normalised MSLP data. The dimensional SAM index produces consistent indices across different resolutions of MSLP data (Fig 3d), that also have consistent spatial correlations with temperature and precipitation (Figs 5 and 6). Although SAM index anomalies are commonly expressed in monthly, seasonal or yearly means, it is the influence of the SAM on synoptic-
scale features such as the path of low pressure system storms and Rossby wave breaking that determines climate impacts (Pepler, 2020; Spensberger et al., 2020). This might suggest that accurate representation of the SAM requires daily or better resolution of MSLP data. However, we demonstrate that the annually averaged climate impacts of the SAM are as effectively represented by latitudinal differences in annual MSLP data as they are for monthly or daily resolution MSLP data (Figs. 5 and 6; A1 and A2), provided that a dimensional SAM index method is used. Correlations of temperature and precipitation
anomalies with the SAM are also consistently stronger for the mid-to-high latitude region where SAM variability is focused when using the dimensional SAM compared with the non-dimensional SAM. This includes an improved representation of the asymmetric (zonal-wave 3) components of SAM variability in the dimensional SAM index, whereas increased weighting of mid-latitude pressure anomalies in the non-dimensional SAM results in increased incorporation of tropical atmospheric circulation anomalies into the SAM index.

Biases in the non-dimensional SAM index appear to be related to the assumed equal weighting of MSLP variability at the mid and high latitudes when the zonal MSLP data is normalised. Instead of assuming either equal (non-dimensional SAM) or no weighting (dimensional SAM) of zonal MSLP data, it could be considered if an equal area weighting based on latitude is optimal for constructing the SAM index. This latitudinal weighting can be achieved by multiplying the zonal MSLP data by the square root of the cosine of latitude (weighting of 0.875 for 40°S and 0.650 for 65°S). This latitudinal weighting has a ratio
of 1.3, which is substantially less than the observed difference in MSLP variability and trends which are approximately 2-3 times larger at 65°S than 40°S. Hence, even when accounting for equal area, the variability and trends in MSLP data remain larger at 65°S and should therefore provide a larger contribution to SAM variability than pressure variability at 40°S (Table A1). This is further verified by repeating our analyses using a dimensional SAM index based on latitude weighted MSLP data. These demonstrate that spatial temperature and precipitation correlations are stronger for the dimensional SAM rather
than a weighted dimensional SAM (Fig. A3-4). The weighted dimensional SAM also has spatial correlation differences when the SAM is calculated at different temporal resolutions which are not present for the dimensional SAM (Fig. A3-4). Hence it appears that area weighting of MSLP anomalies does not improve the representation of the SAM index.

We thus recommend that the best-practice method for calculating the SAM index from zonal MSLP data should be:

$$SAM = P^*_{40°S} - P^*_{65°S} \tag{2}$$

where P*$_{40°S}$ and P*$_{65°S}$ are the zonal MSLP anomalies at 40°S and 65°S, respectively.

Using this method the resulting SAM index will have dimensional pressure units that avoid scaling issues and ambiguity be-tween studies, give appropriate influence to different magnitude of pressure anomalies between the mid-latitudes and Antarc-



tica, produce consistent indices and spatial correlation results across temporal scales, and generate generally stronger relationships to SAM impacts in the southern high latitudes than the traditionally used non-dimensional SAM index.

*Data availability.* All data used in this study are freely available, including by download through Climate Explorer, https://climexp.knmi.nl/start.cgi

*Code availability.* The MATLAB code for data processing and figures is available here: [to be released on Github on publication].

*Author contributions.* N.J.A. conceived the study, L.V.J. carried out data analysis and produced figures. Both authors contributed equally to the discussion of ideas and writing of the manuscript.

*Competing interests.* At least one of the (co-)authors is a member of the editorial board of Climate of the Past.

*Acknowledgements.* This work was supported by the Australian Research Council through a Future Fellowship (FT160100029), Discovery Project (DP220100606) and the Australian Centre for Excellence in Antarctic Science (SR200100008). We thank Sarah Jackson, Georgy Falster and Chiara Holgate for advice and useful discussions that aided in the preparation of this manuscript.



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

## Appendix A: Appendix A

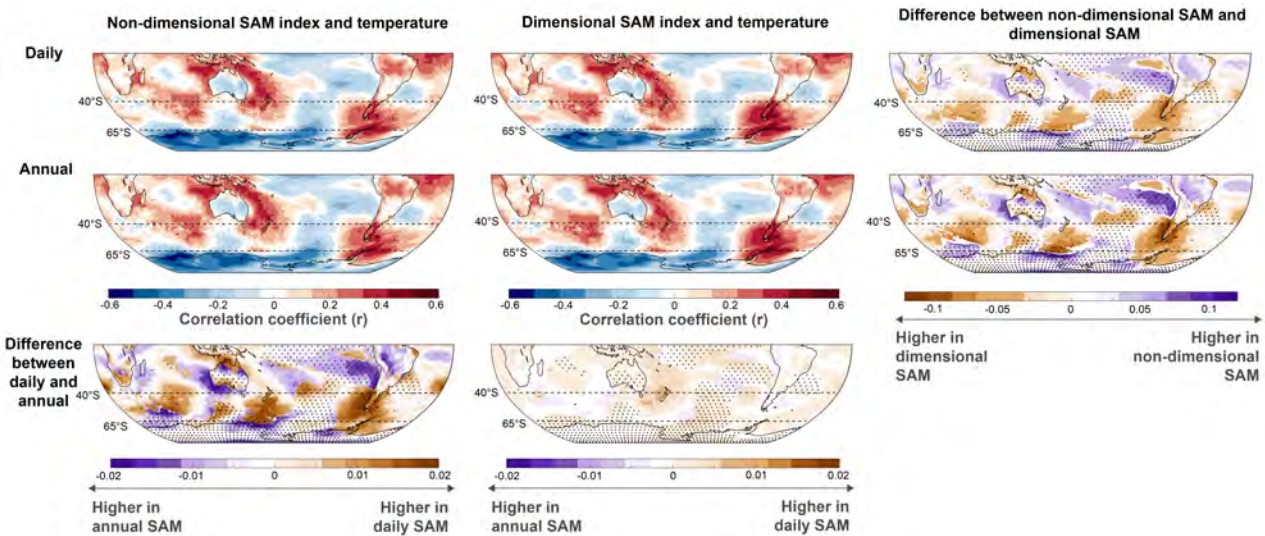

**Figure A1.** Spatial correlation of annual SAM values with ERA5 2m air temperature in the Southern Hemisphere (January-December averages over 1950-2022). Comparisons are shown for differences in SAM indices derived from daily (top row) and annual (middle row) MSLP data, and for non-dimensional SAM indices (left column) and dimensional SAM indices (middle column). Also shown are the differences in spatial correlation values based on MSLP data resolution (bottom row) and for dimensional versus non-dimensional SAM indices (right column). In these correlation difference plots the shading represents differences between methods and data resolution while stippling indicates regions of negative spatial correlations.

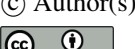



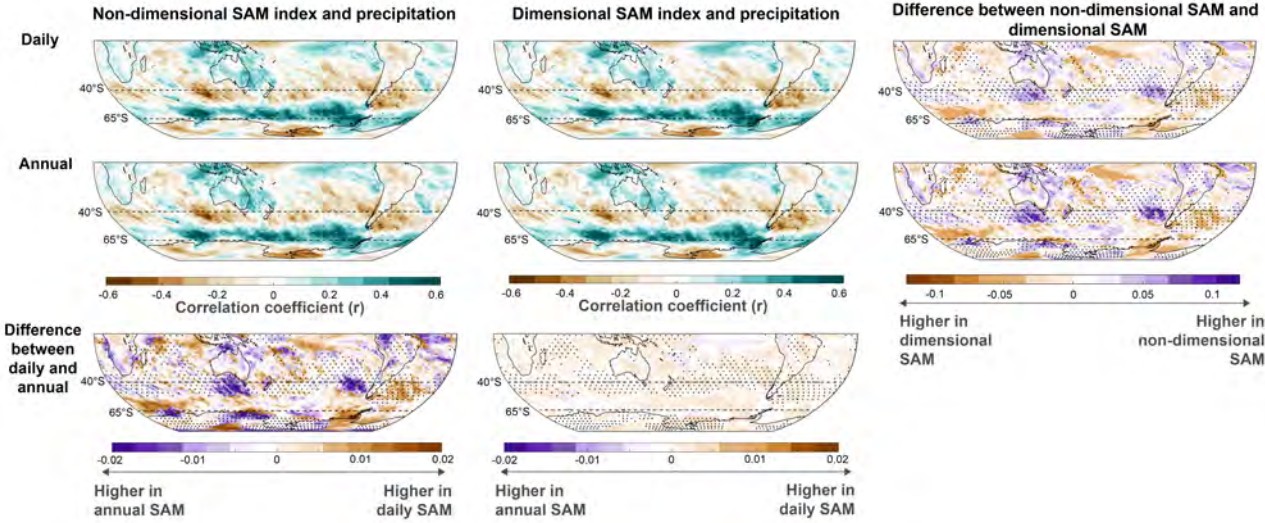

**Figure A2.** Spatial correlation of annual SAM values with ERA5 precipitation in the Southern Hemisphere (January-December averages over 1950-2022). Comparisons are shown for differences in SAM indices derived from daily (top row) and annual (middle row) MSLP data, and for non-dimensional SAM indices (left column) and dimensional SAM indices (middle column). Also shown are the differences in spatial correlation values based on MSLP data resolution (bottom row) and for dimensional versus non-dimensional SAM indices (right column). In these correlation difference plots the shading represents differences between methods and data resolution while stippling indicates regions of negative spatial correlations.

**Table A1.** Characteristics of latitude-weighted MSLP variability and trends for the zonal means used to calculate the SAM index at different data resolutions.

| Data resolution | 40°S standard deviation (1961-1990; hPa) | 65°S standard deviation (1961-1990; hPa) | 40°S trend (1950-2022; hPa/decade) | 65°S trend (1950-2022; hPa/decade) |
|---|---|---|---|---|
| Daily | 2.025 | 3.638 | 0.16 | -0.32 |
| Monthly | 1.482 | 2.616 | 0.16 | -0.32 |
| Annual | 0.446 | 0.881 | 0.16 | -0.32 |



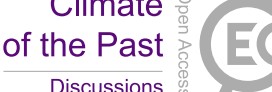

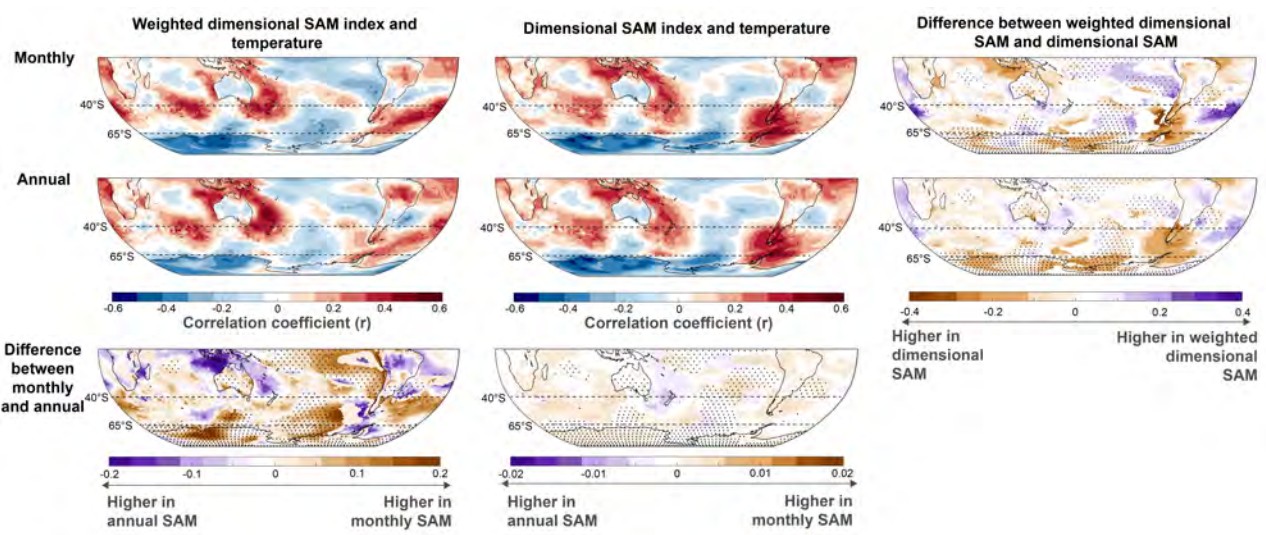

**Figure A3.** Spatial correlation of annual SAM values with ERA5 2m air temperature in the Southern Hemisphere (January-December averages over 1950-2022). Comparisons are shown for differences in SAM indices derived from monthly (top row) and annual (middle row) MSLP data, and for latitudinally weighted dimensional SAM indices (left column) and unweighted dimensional SAM indices (middle column; as in Fig. 5). Also shown are the differences in spatial correlation values based on MSLP data resolution (bottom row) and for dimensional versus non-dimensional SAM indices (right column). In these correlation difference plots the shading represents differences between methods and data resolution while stippling indicates regions of negative spatial correlations.



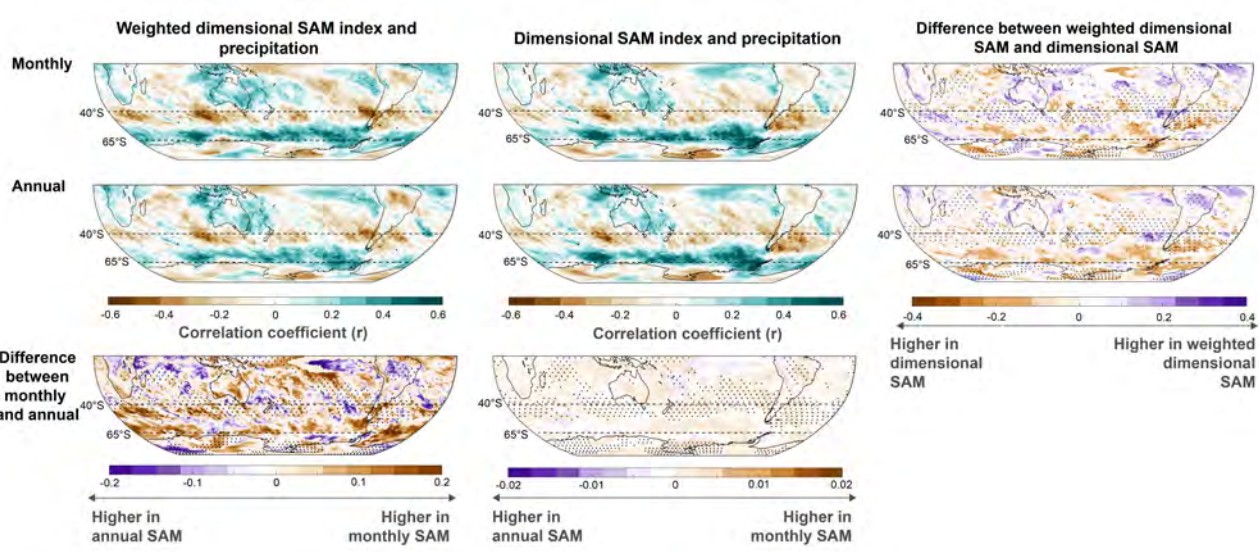

**Figure A4.** Spatial correlation of annual SAM values with ERA5 precipitation in the Southern Hemisphere (January-December averages over 1950-2022). Comparisons are shown for differences in SAM indices derived from monthly (top row) and annual (middle row) MSLP data, and for latitudinally weighted dimensional SAM indices (left column) and unweighted dimensional SAM indices (middle column; as in Fig. 6). Also shown are the differences in spatial correlation values based on MSLP data resolution (bottom row) and for dimensional versus non-dimensional SAM indices (right column). In these correlation difference plots the shading represents differences between methods and data resolution while stippling indicates regions of negative spatial correlations.