# Peer review of "Technical Note: A best-practice approach to calculating the Southern Annular Mode index"

_Climate of the Past, 2023_

## Author Response (AR1)

**Dear Editor and Reviewers,**

**The manuscript has been edited based on the reviewers' comments. We modified the title and the terminology throughout the manuscript. Additionally, we have included supplementary tables and figures that provide more clarity on the relationships between the SAM indices presented in the manuscript, and demonstrate the application of our findings across other versions of the SAM index.**

**Below, we have addressed each of the comments from the reviewers. We want to thank reviewers for their comments that have helped improve the manuscript.**

**Response to comments by Reviewer #1:**

This paper explores how the normalization component of calculating the Southern Hemisphere Annular Mode (SAM) impacts its magnitude and relationships with climate.

I thought that a lot of the manuscript was 'stating the obvious': e.g., the dimensional SAM produces consistent indices across different temporal resolutions. Having said that, no doubt there are many researchers who have correlated something against the SAM without fully understanding how it was produced and how that might impact their findings (e.g., the incorrect Dätwyler methodology). So, in that sense, I think it is a useful addition to the literature, perhaps particularly for the paleoclimate community.

*Response: Thank you for taking the time to review our manuscript. We agree with your sentiments above and this is why we have submitted this as a technical note, rather than a research paper.*

I have a few issues with some of the language used, both the tone and some of the terms used.

One might argue that what the authors are calculating is simply the mean SLP difference between 40°S and 65°S and that this is not the SAM at all. This has been done for other climate indices, such as the NAO, when it is termed the 'natural NAO'. So, maybe this could be referred to as the 'natural SAM' to distinguish it? I agree that in many cases this might be a more useful metric than the SAM (see also Elio Campitelli's comment).

*Response: Thank you for this suggestion. We have changed the terminology throughout the manuscript to normalised SAM index (previously referred to as the non-dimensional SAM index in our manuscript) and natural SAM index (previously referred to as the dimensional SAM index).*

The normalization is done, as it is for many climate indices such as the NAO or SOI, to adjust for seasonal differences in both the average and in the year-to-year range of variability at each of the latitudes, so that each latitude always contributes equally to the index. For certain studies this may be an important criterion. You could argue that the 'natural SAM' is basically a SLP index at the higher latitude, because of the significantly larger variability

there, and maybe if you are interested in how the SAM impacts lower latitudes this is going to swamp the local signal in SLP variability.

*Response: We agree with the reviewer's comments, and believe that this is already covered in our text and figures. Figure 5 and 6 show that the influence of the normalised SAM index versus that natural SAM index emphasises climate impacts in different regions (e.g. the normalised SAM index tends to have stronger correlations in the low latitudes). This is discussed in section 3.2 and in the abstract and conclusions of the paper. We have added additional text to the first paragraph of the discussion, noting that the normalisation allows each latitude to contribute equally to the index.*

I don't like the use of 'best-practice' and 'biases', which comes across as arrogant (how about 'alternative methodology' and 'differences'?): present the positives of your method of producing the 'natural SAM' (which I agree are several) and let the reader decide if they want to use it instead of the standard method. There are also other issues to think about when calculating the SAM, such as using SLP versus say 700-hPa geopotential height as used by the Climate Prediction Center to calculate their SAM/AAO:

(https://www.cpc.ncep.noaa.gov/products/precip/CWlink/daily_ao_index/history/method.shtml).

*Response: We have edited the title and wording in the revised manuscript to reflect this comment. Our revised title is "Technical Note: An improved methodology for calculating the Southern Annular Mode index to aid consistency between climate studies". We have replaced use of the word "biases" in the main text with "differences" or "discrepancies".*

I also concur with Elio Campitelli's comment that a dimensionless SAM is useful when comparing models, which may have differing biases in representing the SH extratropical SLP field.

*Response: We have added a sentence to the discussion to mention that it is possible that normalising the SAM index might somewhat avoid biases between different climate model representations of atmospheric pressure fields in the Southern Hemisphere.*

Specific points

Given that many authors prefer to use an EOF-based SAM index, it would be helpful to do a comparison with this method of deriving the SAM too.

*Response: In our revised manuscript we have calculated a monthly and annual EOF-based SAM index. We found similar differences when using monthly and annual MSLP data to construct the SAM index via an EOF method as what is found through constructing the SAM index using the normalised pressure difference. This has been included as an additional supplementary figure, and additional text added to the results section.*

[Figure]

*Figure A2. a) SAM indices calculated using the EOF method b). Comparison of EOF-derived annual SAM index values derived from monthly versus annual resolution MSLP data. Dashed line represents 1:1 slope.*

I also looked at Fig. 7 and wondered why the differences between the two SAM indices are so much less prior to the reference interval than afterwards. It would be helpful if the authors could investigate this a little more: does it say something about temporal changes in circulation or is it simply biases in the historical model fields?

*Response: Regarding the similarity between SSP1-2.6 and SSP5-8.5 results prior to the reference interval (and indeed prior to 2014), the indices compared across panels a and b are identical from 1850-2014 as they are derived from the same historical simulations.*

*Regarding the normalised and natural SAM indices being more similar prior to 1960 compared to post 1990, this is because the mean SAM index value prior to 1960 is quite similar to that of the reference period, whereas the anthropogenic trends in the SAM become more pronounced in the 21st century allowing the methodological differences between SAM indices to also become more pronounced. This is also why the differences are more pronounced for the high emissions scenario, compared to the low emissions scenario.*

*We have not made any revisions to the text in response to this comment as our analysis is designed only to provide an example of the future climate implications of different versions of the SAM index. To explore the climate processes and implications more fully would require an in-depth multi-model analysis and is beyond the scope of this technical note.*

**Response to comments by Reviewer #2:**

I think this is a good study, as referee 1 already mentioned, in terms of clearing up questions about the obvious. The writing is very concise, and the figure representation is sufficient.

*Response: Thank you for reviewing our manuscript. We appreciate your feedback and have responded to your comments below.*

The questions/comments by Elio Campitelli and referee 1 and the replies by the authors have cleared up most of my questions as well.

As referee1 already mentioned, the authors' method seems to be clearly better when using gridded data, but the reality is that the EOF method is preferred when gridded data is available. It would be nice to see a comparison with an EOF-based index. Also, it's been a while, so I'm not entirely sure, but in a personal conversation with Gong, I remember he said that normalization was done to account for situations where there is not a lot of data, or where there is a lot of uncertainty in the MSLP data, especially where there are only a few in situ observations or paleoclimate proxies.

Would the authors' method be more appropriate for situations where there are only a few observations in each latitude band and the number of samples in each of the two latitude bands is significantly different? Could you discuss this?

*Responses: Our revised manuscript now includes a comparison with the EOF-based method of calculating the SAM index and demonstrates that the normalisation step similarly results in scaling differences based on the temporal resolution of the input data (Figure A2).*

*In the first paragraph of the discussion, we have added "scarcity of observations" to the description of possible reasons why a normalisation step was employed in the original definition of the SAM index.*

*In our revised manuscript we also complement our main analyses with a supplementary figure that also demonstrate the differences between the traditional SAM and the natural SAM when it is calculated using data only from the 12 station locations used for the Mashall SAM index. Using input data corresponding to observational stations (i.e, 40°S= 6 stations, 65°S= 6 stations), we calculated the natural and normalised SAM indices across data resolutions. We found that differences between data resolutions and calculation methods were similar to the results we found when using gridded data (Figure A1).*

[Figure]

*Fig A1. SAM indices calculated using ERA-5 data corresponding to the station locations used for the Marshall observational index.*

*Further to this we also examined whether different numbers of data points in the mid and high latitudes altered our findings. They did not, as the main cause of the differences in the SAM indices is related to the larger magnitude of pressure variability in the high latitude band compared with the mid latitude band. See summary in table below which demonstrates that this characteristic that is seen in zonal mean data is also evident in comparing MSLP variability at mid and high-latitude station observations sites. Hence normalisation does not provide any benefit to overcome issues of a small number of input data when calculating the SAM index. As such we have not included any additional material into our paper around sensitivity testing to small numbers of observational data.*

*Table. Standard deviation of MSLP variability at different resolutions from station locations over the reference period 1960-1991.*

| Station sites | Latitude | Daily (hPa) | Monthly (hPa) | Annual (hPa) |
|---|---|---|---|---|
| Ile Nouvelle Amsterdam | 37°S | 7.70 | 3.26 | 0.95 |
| Gough Island | 40°S | 8.34 | 3.25 | 1.31 |
| Puerto Montt | 41°S | 5.51 | 2.06 | 0.65 |
| Hobart | 42°S | 9.02 | 4.01 | 1.44 |
| Christchurch | 43°S | 9.67 | 4.20 | 1.34 |
| Marion Island | 46°S | 10.33 | 4.14 | 1.08 |
| Faraday | 65°S | 12.31 | 5.65 | 1.66 |
| Mirny | 66°S | 9.58 | 5.11 | 1.75 |
| Casey | 66°S | 10.32 | 5.65 | 1.90 |

| Station sites | Latitude | Daily (hPa) | Monthly (hPa) | Annual (hPa) |
|---|---|---|---|---|
| Dumont D'Urville | 66°S | 10.04 | 5.53 | 1.48 |
| Mawson | 67°S | 9.31 | 4.81 | 1.68 |
| Novolazarevskaya | 70°S | 9.49 | 5.19 | 1.45 |

**Response to comments by Elio Campitelli:**

This is an nice paper. I have to admit that I'd always thought that these indices were always computed using the raw anomalies and share the authors' confusion on why this method is used. (I prefer EOF-based methods). MSLP difference is, to me, the self-evidently physically relevant variable. The authors here also demonstrate empirically other benefits.

*Response: Dear Elio, thank you for taking the time to read our pre-print and provide comments. We appreciate your feedback and have responded below.*

The results in Figure 7 are particularly intriguing, since I would've not expected for the signal-to-noise ratio to change so much.

*Response: Thank you. Yes, while the differences in SAM index methods are small over the observational period, the impact of these differences becomes more evident when looking at longer-term climate change signals.*

I do think that having a dimensionless index is useful, especially for comparing models among themselves and model with observations. So I'd use a dimensionless SAM defined as:

$(P'\_40 - P'\_65)/sd(P'\_40 - P'\_65)$

That is: the (dimensional) difference between MSLP anomalies divided by the standard deviation of such difference computed in the reference period. This index is essentially the same as the dimensional index in this paper but scaled by a constant factor, so it should share all its good properties of plus it has the advantage of being easy to compare between models.

*Response: Thank you for this suggestion. We calculated the SAM index using the method you suggested. We found that this method also led to differences between data resolutions.*

*Although the method suggested would provide a way of standardising the SAM index produced between different climate models, the normalisation step is still subject the differences in variability based on the resolution of the data being used and so the same issues in generating a standardised SAM index still emerge. As noted in the start of the reviewer's comments it is the actual pressure difference that is the physically relevant variable and this standardisation step would obscure actual differences in the representation of the SAM between different climate models.*

Some comments below:

Figure 3b shows that the correlation of the annual non-dimensional SAM indices is essentially 1. A correlation of 0.9998 between indices is basically perfect and is much greater than the correlation of SAM indices computed using different datasets, different reference periods, different latitude bands, different levels, or slightly different methods. Therefore, they are basically equivalent except for a constant scaling factor (shown as the slope). Being dimensionless quantities, this scaling factor is not relevant. Particularly, it's not relevant for correlations.

*Response: Yes, the difference in the non-dimensional SAM index calculated different resolution is most evident in the scaling factor. You are correct that this should not be particularly relevant for a dimensionless quantity, but it is an issue that has resulted in confusion across previous literature. For example, for many years it was thought that there were large differences in the magnitude of SAM changes reconstructed over the last millennium. This can be traced back to the resolution of instrumental MSLP data used to calculated the observational SAM index that is used as the target for the paleoclimate reconstruction. Because the indices produced are dimensionless it is very hard to trace that this is where the discrepancy between reconstructed values originates from. Our study (Figure 3d) shows that dimensional SAM index that retains pressure units avoids introducing any scaling factors that in the past have resulted in ambiguities between SAM studies.*

*We also demonstrate that while the difference between the normalised SAM indices is the scaling factor, there are additional small differences which are relevant for spatial correlations that are introduced by the normalised SAM giving equal weighting to MSLP changes in the mid and high latitudes (Figure 5). Hence, we do not agree with the statement that differences in the SAM indices are not relevant for correlations.*

This can be seen in the correlation differences shown in the bottom left panel of Figures 5 and 6, which are tiny (on the order of 0.01). These differences are probably not statistically significant and, IMHO, most definitely not physically significant. They are likely much smaller than the standard error of their estimations or the difference between various reasonable SAM indices.

*Response: We agree that the differences in spatial correlations from different resolutions of the SAM index (bottom row of Figures 5 and 6) are small and over the observational period will not be physically significant. However, our point here is not to assess significance, it is merely to show that there are some differences in spatial correlations that are introduced solely by data resolution when using a normalised SAM index. In contrast, these resolution-based differences are essentially completely avoided when using a natural SAM index.*

*More importantly in Figure 5 and 6 we demonstrate that there are much larger differences in spatial correlations when using a natural or normalised SAM index (far right column of Figures 5 and 6). This is due to the normalised SAM index giving an equal weighting to*

*pressure changes at the mid and high latitudes, when in reality the MSLP variability and trends at the high latitudes have a larger magnitude than those at the mid latitudes. In this case the differences in correlation values are on the order of 0.1, which will influence interpretations of statistical significance in some regions.*

Figure 3d: Why are not the three time series exactly equal? The monthly mean of the difference between daily values must be exactly the same as the difference between monthly mean values. If they are not equal, what's their correlation?

*Response: The natural SAM indices are not exactly equal. This is a result of averaging the MSLP data at different resolutions. For the daily SAM index each of the 365/366 days in the year has equal weighting when then averaging to an annual SAM value. For the monthly SAM index the different numbers of days per month gives very slightly different weighting to how the data for each day contributes to the calculated annual SAM value.*

Figure 4: What's the correlation between the dimensional and non-dimensional indices at the annual, monthly and daily resolution? It would be helpful to have a number attached to these differences.

*Response: We added a supplementary table (Table A1) with the correlation coefficients and slope values for the different data resolutions between calculation methods.*

Figure 5: There's a strange artefact in the top-left panel near the dateline. A vertical discontinuity in the shading. The artefact might also be present in the other panels but harder to see.

*Response: Thank you for spotting this. After inspecting the maps, we have found that the artefact comes from the preprint and not from the figure itself. This should not appear once the manuscript is publish using the high-resolution figures.*

Figure 5 and 6: Would it be better to plot the difference in r2 between each panel? Right now, to interpret the figure one needs to look also at the sign of the correlation in the original panels. By plotting the difference in r2, negative values would be directly interpretable as a decrease in the strength of the (linear) relationship and vice versa. Are the differences in correlation statistically significant?

*Response: This interpretation of what we have plotted isn't correct. We have already made this adjustment for correlation sign in calculating the correlation differences, but we will clarify this in our figure caption text. The process involved turning the correlation values into absolute values so that the difference between the correlations could be used to determine which version of the SAM index resulted in stronger correlations (positive values are stronger in one version, negative values are stronger in the other version). This is what is shown by the shading on the difference plots. The stippling on the difference plots is used to refer back to where correlations values were negative prior to being made*

*into absolute values. For grid cells where the correlation values were a different sign between the two versions of the SAM, these correlation values were very close to zero, and so were set to zero for plotting the correlation differences.*

*Since we are not testing a hypothesis in our study, we did not conduct a significance analysis. We are simply demonstrating that differences in data resolutions and calculation methods can carry over into subsequent analysis (such as spatial correlations). While these differences are small over the historical era, they become more consequential as the climate change influence on the SAM becomes stronger this century (Figure 7).*

Figure 4 and 7: How are these plots created? By definition, the units of the dimensional and non-dimensional indices are not the same so their alignment should be arbitrary (in other words, the y axis has two different units). Did you standardise both indices to show them on the same scale?

*Response: We have edited figures 4 and 7 to account for the different scales between SAM indices. In case of Figure 4, we have scaled the Y axis relative to the slope between both time series. Regarding Figure 7, we have scaled the y axis relative to the regression over the reference interval. This scaling description has been included in the legend of each figure.*

Figure 7: It's surprising, at least to me, that both indices are almost exactly the same before ~1950. Why is that? Was there a change in the ratio of the standard deviation of MSLP between 40ºS and 60ºS?

*Response: We weren't sure if you were referring to the similarities between the natural and normalised SAM indices before 1950, or the identical SAM indices between panel a and b? Both panels use the same historical simulation so panels a and b are identical until 2014 when the SSP future scenarios begin. The very close similarities between the natural and normalised SAM indices before 1950 are consistent with what we show in Figure 4, where the historical differences are small (but not altogether negligible in some years). As the climate change influence on the SAM develops during 20th and 21st centuries the differences between the natural and normalised SAM indices increases.*